# An Improved Spatiotemporal Data Fusion Method for Snow-Covered Mountain Areas Using Snow Index and Elevation Information

**DOI:** 10.3390/s22218524

**Published:** 2022-11-05

**Authors:** Min Gao, Xingfa Gu, Yan Liu, Yulin Zhan, Xiangqin Wei, Haidong Yu, Man Liang, Chenyang Weng, Yaozong Ding

**Affiliations:** 1Aerospace Information Research Institute, Chinese Academy of Sciences, Beijing 100049, China; 2University of Chinese Academy of Sciences, Beijing 100049, China; 3School of Remote Sensing and Information Engineering, North China Institute of Aerospace Engineering, Langfang 065000, China

**Keywords:** remote sensing, spatiotemporal data fusion, ESTARFM, NDSI, DEM

## Abstract

Remote sensing images with high spatial and temporal resolution in snow-covered areas are important for forecasting avalanches and studying the local weather. However, it is difficult to obtain images with high spatial and temporal resolution by a single sensor due to the limitations of technology and atmospheric conditions. The enhanced spatial and temporal adaptive reflectance fusion model (ESTARFM) can fill in the time-series gap of remote sensing images, and it is widely used in spatiotemporal fusion. However, this method cannot accurately predict the change when there is a change in surface types. For example, a snow-covered surface will be revealed as the snow melts, or the surface will be covered with snow as snow falls. These sudden changes in surface type may not be predicted by this method. Thus, this study develops an improved spatiotemporal method ESTARFM (iESTARFM) for the snow-covered mountain areas in Nepal by introducing NDSI and DEM information to simulate the snow-covered change to improve the accuracy of selecting similar pixels. Firstly, the change in snow cover is simulated according to NDSI and DEM. Then, similar pixels are selected according to the change in snow cover. Finally, NDSI is added to calculate the weights to predict the pixels at the target time. Experimental results show that iESTARFM can reduce the bright abnormal patches in the land area compared to ESTARFM. For spectral accuracy, iESTARFM performs better than ESTARFM with the root mean square error (RMSE) being reduced by 0.017, the correlation coefficient (r) being increased by 0.013, and the Structural Similarity Index Measure (SSIM) being increased by 0.013. For spatial accuracy, iESTARFM can generate clearer textures, with Robert’s edge (Edge) being reduced by 0.026. These results indicate that iESTARFM can obtain higher prediction results and maintain more spatial details, which can be used to generate dense time series images for snow-covered mountain areas.

## 1. Introduction

Snow cover is an important component of the cryosphere and has significant effects on regional water balance, local weather, atmospheric circulation, and surface hydrological processes [1,2,3,4]. Previous studies have shown that small changes in snow cover in mountain areas may have great thermal and dynamical influences on regional and even global circulation systems [5,6,7,8], which is an essential factor of energy balance. Remote sensing images with fine spatial and temporal resolutions contain much valuable information about the observed objects [9,10,11,12], which is a fundamental source for studying and monitoring the spatiotemporal distribution of snow cover [13,14,15]. However, due to technical and budget limitations, there is a trade-off between the swath width and the revisit cycle of satellites. So, it is difficult to obtain images with both high spatial and temporal resolution by a single satellite [16], especially in the mountain region, where the spatial and temporal variability of the snow cover is particularly high [17]. Furthermore, optical remote sensing images are easily contaminated by clouds, cloud shadows, and atmospheric conditions, especially in cloudy and snowy areas such as tropical, subtropical, and high-altitude mountain areas [18]. As a result, the present remote sensing datasets cannot satisfy the need for high spatial time series observation.

The spatiotemporal data fusion method (STF) is a flexible, effective, and inexpensive solution to overcome these limitations. STF aims at fusing images with low spatial high temporal resolution and images with high spatial low temporal resolutions to generate images with high spatial and temporal resolution [19]. In the past decade, this method has been applied to a variety of research fields, such as crop monitoring, land cover classification, biomass estimation, and disturbance detection [20,21,22,23,24]. Meanwhile, a large number of STF methods have been developed in the remote sensing field. According to their principles, assumptions, and strategies, the STF methods can be divided into five categories [25,26,27,28]: weight function-based [29,30], unmixing-based [31,32], Bayesian-based [33], learning-based [34,35], and hybrid method [36,37]. Among these methods, weight-based methods are most widely applied in practical research because of their high prediction accuracy, good robustness, and flexibility [38,39,40,41]. The enhanced spatial and temporal adaptive reflectance fusion model (ESTARFM) was developed, aiming to improve applicability in heterogeneous regions [42]. This method is based on two pairs of fine and coarse images at base date and a coarse image at target time, which can obtain more spatial and temporal information. The ESTARFM can be reduced to three steps. Firstly, select similar pixels. The selection of similar pixels is based on the thresholds, which are estimated from the overall standard deviation of the whole image. Secondly, calculate weights and conversion coefficient. The weights are composed of spectral, temporal, and spatial distances. Finally, calculate the reflectance of the target pixel. Some experimental results show that ESTARFM can maintain fine spatial and spectral details as well as obtain high prediction accuracy in heterogeneous regions, such as agricultural areas and urban, coalfields [43,44,45,46]. However, ESTARFM still has several limitations and constraints in practical applications [36,40]. There are many studies that aim to solve the limitations in different fields. In order to overcome the uncertainty in selecting neighboring similar pixels, an improved ESTARFM method was developed based on the landcover endmember type [47]. To improve the prediction accuracy in non-shape-changing regions, Zhang et al. proposed an object-based method [48]. Aiming to improve the prediction accuracy of ESTARFM, a statistical method was proposed to adaptively determine the window size [42]. In order to improve the selection accuracy of similar pixels for the Paddy rice region, an enhanced vegetation index (EVI) was introduced for predicting reflectance [49]. To make it applicable to larger scales, an automatic filling cloud gaps framework was proposed to enhance the applicability in heterogeneous and cloud-prone landscapes [41]. Although these methods can obtain good prediction accuracy in the specific study areas, there is still a lack of specific methods in the snow-covered mountain regions where transient changes often occur due to snow and snowmelt [50].

There are some limitations of ESTARFM when applied to snow-covered mountain areas [51,52]. First, snow reflectance is higher than non-snow pixels in the visible band, which leads to a high threshold of selecting similar pixels, resulting in large errors. Similarly, for non-snow pixels, it could lead to a low threshold. Secondly, the type of pixels would change due to snow and snowmelt, while ESTARFM selects similar pixels based on the intersection of two fine images, which also leads to the error of selecting similar pixels. The above problems will lead to a decrease in prediction accuracy. Therefore, it is important to accurately capture snow-cover changes to improve prediction accuracy. The normalized snow index (NDSI) takes advantage of the characteristics of snow, which has a high reflection in the visible bands and strong absorption in the shortwave infrared, so a suitable threshold can distinguish snow and non-snow pixels [50,53]. For mountain areas, the variation in snow has a high correlation with elevation. So, the digital elevation model (DEM) is essential for the analysis of the snow and glacier change in high mountain terrains [54]. Additionally, several studies have shown the applicability of morphometric parameters derived from DEM in the mapping of glaciers and snow cover [54,55,56,57,58,59]. Thus, NDSI and DEM are selected for estimating the change in snow.

This study proposes an improved ESTARFM (iESTARFM) for snow-covered mountain areas in the Nepal region by introducing NDSI and DEM information. Compared to the original ESTARFM, there are three improvements. Firstly, simulate snow-cover changes for the base time images using NDSI by taking advantage of the reflectance characteristics of snow. DEM data was introduced to simulate snow-cover for the target time images because of the high correlation between snowmelt and elevation. Secondly, select similar pixels. Similar pixels are selected according to the results of snow-covered changes. If the pixels at the target date are identified as snow, then similar pixels will be selected from the snow pixels in the window size. The same principle is applied to non-snow pixels. The thresholds are calculated separately according to the pixel type. Thirdly, calculate weights and target pixels. To reduce the error caused by the misclassification, NDSI is added to select similar pixels and calculate the weights; more similar pixels have more similar NDSI values and are assigned a larger weight value. Finally, the target pixels are calculated according to the similar pixels and their weights. The data used in this study are Landsat 8 surface reflectance images with high spatial resolution and MODIS surface reflectance products (MOD09A1) with a high temporal resolution, which are widely used in spatiotemporal data fusion methods [19,30,38,58]. The rest of the paper is organized as follows. Section 2 introduces the study area and datasets. Section 3 describes the details of the proposed iESTARFM method. Section 4 evaluates the performance of iESTARFM and compares it to the original ESTARFM. Section 5 and Section 6 discuss and conclude the advantages and limitations of our method.

## 2. Materials

### 2.1. Study Area

Nepal experiences a wide range of climatic conditions that can be divided into two types: a dry winter period and a wet summer period. There are six bioclimatic zones that vary greatly in this climate, and they are tropical, subtropical, temperate, subalpine, alpine, and nival types [60]. The northern part of Nepal is a mountainous country that covers two-thirds of the Himalayan region with the eight highest mountains in the world [61]. Snow cover is one of the major types in Nepal [62], and there is seasonal snowfall in the region. The study area is located in the northwest part of Nepal in the mountain area, which is adjacent to China and the Himalayas, and the main land cover in this area is grass and snow. Snow cover changes rapidly in this study area, making the area suitable for testing the proposed method. Figure 1 shows the location of the study area and the Landsat image on 12 February 2020 using an RGB composite.

### 2.2. Satellite Data and Preprocessing

The experiment data include Landsat 8 surface reflectance (SR) products and MODIS surface reflectance products (MOD09A1). Because of the high spatial resolution of Landsat and the high temporal resolution of MODIS, they are widely used in spatiotemporal data fusion methods [19,30,38,63]. The datasets used in this paper were downloaded from Google Earth Engine (GEE). The Landsat 8 SR product contains five visible and near-infrared bands and two short-wave infrared (SWIR) bands. The dataset was processed to orthorectified surface reflectance with a spatial resolution of 30 m [64]. This product was generated using the Land Surface Reflectance Code (LaSRC). The MODIS surface reflectance product (MOD09A1) provides MODIS surface reflectance of seven bands at a resolution of 500 m, which was selected from the best L2G observations during an 8-day period [65]. The band information of Landsat and MODIS is shown in Table 1. The Advanced Spaceborne Thermal Emission and Reflection Radiometer global digital elevation model (ASTER GDEM) data were selected as digital elevation model (DEM) data. ASTER GDEM provides a higher spatial resolution of 30 m and wider land coverage for estimating the extent of snow cover on the target dates [66].

The input data is the same as required by ESTARFM [29]. Our experiment requires two pairs of Landsat and MODIS SR images at the same date as well as a MODIS image at the target date as base images. A set of images with snow-cover changes from January to March were selected. Considering that the MODIS image is an eight-day composite product, the MODIS and Landsat images with the closest dates were selected. The images in January ($t_{1}$) and March ($t_{2}$) were used to predict the image in February ($t_{p}$). The data used in the experiment are listed in Table 2. The Landsat data in February were used as comparison data for accuracy verification. The cloud coverage of all the images is less than 5%, and the missing values were filled in with ENVI. The percentage of snow cover was calculated using the threshold method with NDSI > 0.4. Figure 2 shows the MODIS and Landsat surface reflectance images used in our experiment. It can be seen that the snow cover gradually decreases from January to March. MODIS images were resampled to the same spatial resolution of 30 m as Landsat. All the images were collected and cut to the same extent as the study area.

## 3. Methods

### 3.1. Description of the Improved ESTARFM

To describe the method more clearly, we explain some definitions in advance. Our experiment requires two pairs of Landsat and MODIS SR images at the same date as well as a MODIS image at the target date as base images. Landsat in January, February, and March are marked as $F_{t_{1}}$, $F_{t_{p}}$, and $F_{t_{2}}$. Meanwhile, MODIS in January, February, and March are marked as $C_{t_{1}}$, $C_{t_{p}}$, and $C_{t_{2}}$.

The flow chart of the improved ESTARFM (iESTARFM) is shown in Figure 3. Compared to the original ESTARFM, there are three improvements for snow-covered mountain regions in iESTARFM [42]. First, snow cover changes are simulated based on NDSI and DEM data by taking advantage of the spectral characteristics that snow has a high reflection in the visible bands and strong absorption in the shortwave infrared. A suitable threshold can distinguish between snow and non-snow pixels [50,53], and the details are shown in Section 3.2. DEM is important for estimating the volume change for inaccessible snow-covered mountain regions [9,13,14]. Considering that the variation in snow-covered mountain areas has a high correlation with elevation, it is feasible to simulate the snow cover based on a suitable DEM threshold, and the details are shown in Section 3.3. Second, similar pixels are selected according to the results of snow cover changes simulated by NDSI and DEM. Thirdly, the thresholds of selecting similar pixels are calculated separately according to snow and non-snow pixels. To reduce the error caused by the misclassification, NDSI is added to select similar pixels and calculate the weights. The detailed descriptions of iESTARFM are given below. For more ESTARFM information please refer to [42].
(1)Simulate snow cover based on NDSI at the base date. The normalized difference snow index (NDSI) is widely used for snow identification by taking advantage of the spectral characteristics that snow has high reflectance in the green band and low reflectance in the short-wave infrared band. Based on this, the ratio of the two bands is calculated to highlight the characteristics of snow from others [53], and the calculation equation is as follows:
(1)$$NDSI=\frac{\rho_{green}-\rho_{swir2}}{\rho_{green}+\rho_{swir2}}$$
where $\rho_{green}$ is the reflectance of the green band, $\rho_{swir2}$ is the reflectance of the SWIR2 band.Firstly, by analyzing the NDSI distribution histogram and the true surface reflectance, the frequency histogram has two peaks, where the one with high values indicates the snow area and the one with low values indicates the non-snow area. Secondly, the NDSI threshold was determined by experimenting with different NDSI threshold values between the two peaks. Thirdly, the pixels with NDSI smaller than the threshold are considered as non-snow pixels and marked as 0, and those with NDSI larger than the threshold are considered as snow pixels and marked as 1. The experimental details are shown in Section 3.2, and the equation is as follows:(2)$$\begin{array}{cc} \mathrm{If}\;|NDSI_{F}\left(x_{i},y_{i},t_{k}\right)|\leq \sigma_{snow} & Mask_{NDSI}\left(x_{i},\;y_{i},t_{k}\right)=0\\ \mathrm{else} & Mask_{NDSI}\left(x_{i},y_{i},t_{k}\right)=1 \end{array}$$
where $\sigma_{snow}$ is the threshold to mask the snow-covered map, ($x_{i},y_{i})$ is the coordinate of the ith pixel, t is the base date, $t_{k}$ can be either $t_{1}$ or $t_{2}$, $Mask_{NDSI}\left(x_{i},\;y_{i},t_{k}\right)$ is the snow-covered mask, with 1 indicating snow pixels and 0 indicating non-snow pixels.
(2)Simulate snow cover based on DEM at the target date. The elevation is an important factor affecting the spatiotemporal distribution of the snow cover in mountainous areas [54,55,56,57,58,59]. The temperature at high altitudes is low, which is suitable for snow accumulation; meanwhile, the relatively high temperatures at lower altitudes cause snow to melt faster. The distribution of snow is strongly correlated with the elevation. Therefore, the combination of NDSI products and DEM is an effective approach for studying the spatial and temporal distribution of snow in mountain areas. Firstly, the coarse snow-covered borders are extracted from MODIS NDSI by using the local binary pattern (LBP) operator [67]. Then, the DEM values located at the boundaries are counted, and the threshold value of DEM is obtained. Thirdly, the pixels with DEM smaller than the threshold are considered non-snow pixels and marked as 0, and those with DEM larger than the threshold are considered snow pixels and marked as 1. The details are shown in Section 3.3, and the equation is as follows:
(3)$$\begin{array}{cc} \mathrm{If}\;DEM\left(x_{i},y_{i}\right)\leq \sigma_{DEM} & Mask_{DEM}\left(x_{i},\;y_{i},t_{p}\right)=0\\ \mathrm{else} & Mask_{DEM}\left(x_{i},\;y_{i},t_{p}\right)=1 \end{array}$$
where $\sigma_{DEM}$ is the threshold of DEM to classify snow and non-snow pixels, $Mask_{DEM}\left(x_{i},\;y_{i},t_{p}\right)$ is the mask obtained by thresholding at the target date, with 1 indicating snow pixels and 0 indicating non-snow pixels.
(3)Select similar pixels. ESTARFM selects similar pixels based on spectral similarity [42]. The threshold is determined by the standard deviation of a population of pixels from the base image with a high spatial resolution. The improved method differs from ESTARFM in that it does not calculate thresholds based on the whole image but calculates the threshold for the snow and non-snow pixels separately to reduce errors in selecting similar pixels. Meanwhile, more similar pixels have more similar NDSI values, so NDSI is used as an additional condition to improve the accuracy of selecting similar pixels. Finally, similar pixels are selected based on spectral and NDSI differences. The equation is as follows:(4)$$\left|F\left(x_{i},y_{i},t_{k},B\right)-F\left(x_{w/2},y_{w/2},t_{k},B\right)\right|\leq \sigma_{flag}\left(B\right)\times 2/m$$
(5)$$\left|NDSI\left(x_{i},y_{i},t_{k}\right)-NDSI\left(x_{w/2},y_{w/2},t_{k}\right)\right|\leq \sigma_{NDSI}$$
where F is the spectral reflectance of fine images, w is the size of the searching window, σ(B) is the standard deviation of reflectance for band B, m is the estimated number of classes. Flag means the pixels marked as snow or non-snow, with flag = 1 indicating snow pixels and flag = 0 indicating non-snow pixels. $\sigma_{flag}\left(B\right)$ is the threshold for non-snow or snow pixels, and $\sigma_{NDSI}$ is the threshold based on NDSI standard deviation.(4)Calculate weights and the conversion coefficient. The weight calculation in ESTARFM involves spectral, temporal, and spatial distances. Similarly, in iESTARFM, the spectral distance is calculated using the correlation coefficient between the fine image and the coarse image at $t_{k}$. The spatial distance is calculated using the geographic distance between the similar pixels and target pixels. The temporal distance is calculated as the spectral difference between two coarse images. The conversion coefficient is the ratio of the change in the pure pixels in the fine image to the change in pixels in the coarse image, and it is calculated by a linear regression model. iESTARFM adds NDSI to the weight calculation to reduce the error of snow and non-snow identification. The weights equation is defined as follows:(6)$$R_{i}=\frac{E\left[\left(F_{i}-E\left(F_{i}\right)\right)\left(C_{i}-E\left(C_{i}\right)\right)\right]}{\sqrt{D\left(F_{i}\right)}\sqrt{D\left(C_{i}\right)}}$$
(7)$$d_{i}=1+\sqrt{{\left(x_{w/2}-x_{i}\right)}^{2}+{\left(y_{w/2}-y_{i}\right)}^{2}}/\left(w/2\right)$$
(8)$$NDSI_{i}=\left|NDSI\left(x_{i},y_{i},t_{k}\right)-NDSI\left(x_{w/2},y_{w/2},t_{k}\right)\right|+1$$
(9)$$D_{i}=\left(1-R_{i}\right)\times d_{i}\times NDSI_{i}$$
(10)$$W_{i}=\left(1/D_{i}\right)/\overset{N}{\underset{i=1}{\sum }}\left(1/D_{i}\right)$$
where $R_{i}$ is the spectral correlation coefficient for the ith pixel, $d_{i}$ is the spatial distance, E is the expected value, $NDSI_{i}$ is the spatial distance, D is the variance, $d_{i}$ is the spatial distance, $D_{i}$ is the normalized reciprocal of weight, and $W_{i}$ is the normalized weight.(5)Predict the target pixels. The prediction of the target pixels can be divided into two cases. Case 1: the pixels are marked with the same type at $t_{1}$,$\;t_{2}$, and $t_{p}$, indicating the type has not changed, so the target pixels can be calculated based on $t_{1}$ and $t_{2}$. Case 2: the pixels are marked with the same type only on one date as $t_{p}$, indicating that there is a change, so the target pixels are predicted based on the pixels marked with the same type. The equation is as follows:(11)$$\mathrm{CASE}1:Mask_{NDSI}\left(x_{i},\;y_{i},t_{1}\right)=Mask_{NDSI}\left(x_{i},\;y_{i},t_{2}\right)=Mask_{DEM}\left(x_{i},\;y_{i},t_{p}\right)$$
(12)$$F\left(x_{w/2},y_{w/2},t_{p},B\right)=T_{1}\times F_{1}\left(x_{w/2},y_{w/2},t_{p},B\right)+T_{2}\times F_{2}\left(x_{w/2},y_{w/2},t_{p},B\right)$$
(13)$$\mathrm{CASE}2:\;Mask_{NDSI}\left(x_{i},\;y_{i},t_{k}\right)=Mask_{DEM}\left(x_{i},\;y_{i},t_{p}\right)$$
(14)$$\;\;\;\;\;\;\;\;\;\;\;F\left(x_{w/2},y_{w/2},t_{p},B\right)=F\left(x_{w/2},y_{w/2},t_{k},B\right)+\overset{N}{\underset{i=1}{\sum }}W_{i}\times V_{i}\times \left(C\left(x_{i},y_{i},t_{P},B\right)-C\left(x_{i},y_{i},t_{K},B\right)\right)$$
where $Mask_{NDSI}\left(x_{i},\;y_{i},t_{k}\right)$ is the snow-covered mask obtained by calculating the NDSI threshold, and $Mask_{DEM}\left(x_{i},\;y_{i},t_{p}\right)$ is the snow-covered mask obtained by calculating the DEM threshold. $F\left(x_{w/2},y_{w/2},t_{p},B\right)$ is the predicted surface reflectance of the target pixels, $F_{1}$ is the prediction result calculated from the base image at time $t_{1}$, and $F_{2}$ is the prediction result calculated from the base image at time $t_{2}$.

### 3.2. Simulate Snow Cover Based on NDSI at the Base Date

The normalized snow index (NDSI) takes advantage of snow’s high reflection in the visible bands and strong absorption in the shortwave infrared, so the selection of a suitable threshold can distinguish snow from other components [50,53]. When the NDSI of the pixels is greater than the threshold, the pixels are marked as snow; otherwise, they are marked as non-snow. In this study, five base images were selected for the experiment, including a pair of Landsat and MODIS images in January and March and a MODIS image in February, as shown in the first row of Figure 4a–e. The NDSI images corresponding to the five base images are shown in the second row of Figure 4f–j. The histogram of the distribution of NDSI values is shown in the third row of Figure 4k–o. By analyzing the histogram and surface reflectance images, it was found that the NDSI frequency distribution histogram has two peaks, where the peak of the low NDSI indicates the distribution of non-snow pixels and the peak of the high NDSI indicates the distribution of snow pixels. It can be found that the NDSI thresholds for snow-covered mapping are distributed between [−0.2, 0.5] for MODIS images and [0, 0.5] for Landsat images. Meanwhile, by analyzing the NDSI distribution histogram, the values that can distinguish between these two peaks are distributed in the range [0, 0.5]. Then, by experimenting with different NDSI threshold values, it was found that the NDSI threshold of 0.4 can distinguish snow and non-snow pixels well. Thus, the NDSI threshold of 0.4 was chosen to calculate the snow-covered mask, as shown in the fourth row of Figure 4p–t.

### 3.3. Simulate Snow Cover Based on DEM at the Target Date

To improve the prediction accuracy, it is necessary to obtain the type of pixels at the target time. However, the target data does not have a high-resolution image but only the MODIS low spatial resolution image. DEM is essential for analyzing the snow change in high mountain terrains [54]. As shown in Figure 5a–c are the true Landsat surface reflectance images on 2020/01/11, 2020/02/12, and 2020/03/31. There is an obvious change in snow melting from January to March. Figure 5d–f are the corresponding snow masks by setting different thresholds. It can be seen that there is a strong relationship between DEM and the snow cover boundary.

Considering that the variation in snow cover in mountainous areas has a high correlation with elevation, it is feasible to simulate the snow cover at the target time based on DEM data. Firstly, snow cover boundaries are extracted by using the coarse-resolution MODIS NDSI at the target time. The local binary pattern (LBP) operator is robust and computationally simple for texture analysis, and it applies the statistical and structural approach to texture analysis [68]. The LBP operator labels 3 × 3 neighborhood pixels for each central pixel of the image with a binary number by comparing gray values between the central pixel and each neighborhood pixel. In this way, the LBP operator can describe the structural information, thus providing excellent texture extraction for NDSI binary masks. The LBP operator is applied to the MODIS NDSI at the target time to extract the boundary of the snow-covered area, and the result is shown in Figure 6a. The equation is as follows:(15)$$LBP\left(x_{c},\;y_{c}\right)=\sum_{P=0}^{P-1}2^{P}s\left(i_{p}-i_{c}\right)$$
(16)$$\mathrm{where}\;s\left(x\right)=\{\begin{array}{c} 1\;if\;x\geq 0\\ 0\;if\;x<0 \end{array}$$
where *P* is the *p*th pixel in the 3 × 3 window, $i_{p}$ is the grayscale value of the *p*th pixel, $i_{c}$ is the grayscale value of the central pixel.

Secondly, a statistical analysis is performed on the DEM data located at the boundary. The frequency histogram is shown in Figure 6b. It can be seen from the frequency histogram that the DEM height is mostly distributed in the range [3590, 3630]. So, the boundary extracted from MODIS NDSI is overlaid on the DEM height of 3600 m, as shown in Figure 6c. Finally, the DEM at a height of 3600 m is used as the threshold to extract the snow cover at the target time. To avoid misclassification of pixels at the boundary, this paper chooses a buffer of 500 m (one MODIS pixel). The snow-covered mask is shown in Figure 6d.

### 3.4. Data Quality Evaluation Metrics

Satellite images mainly contain spectral and spatial information. So, to quantitatively evaluate the accuracy of the proposed method, eight accuracy evaluation metrics including spectral and spatial evaluation metrics [27,69] were adopted to compare the predicted images with the true images [70]. The spectral accuracy metrics include mean square error (MAE), root mean square error (RMSE), the Pearson correlation coefficient (r), relative global dimensional synthesis error (ERGAS), Structural Similarity Index Measure (SSIM), Spectral Angle Mapper (SAM) and Peak Signal-to-Noise Ratio (PSNR). MAE and RMSE have similar meanings in the accuracy evaluation and are usually used to measure the difference between the predicted images and the true images. For MAE and RMSE, the value closer to 0 means that the predicted result is more similar to the real image, indicating a more accurate predicted result, while a larger value means that the predicted image deviates more from the real image. The Pearson correlation coefficient (r) indicates the linear relationship between the predicted image and the true image, and a value closer to 1 indicates a better correlation between the predicted and true values. ERGAS is used to evaluate the overall fusion result, and a value closer to zero indicates higher overall fidelity of the predicted image [71]. SSIM is an evaluation metric often used in computer vision to measure image similarity [72], which evaluates images in terms of luminance, contrast, and structure. In this study, this metric is used to evaluate the overall structural similarity of the image. A value closer to 1 indicates that the two images are more similar, and the larger the value, the better the image quality. SAM measures the spectral distortion of the fusion result. The smaller the value, the closer the image to the real image [73]. PSNR can evaluate the quality of the predicted result, and a higher value indicates that the predicted image has a better quality [74]. Spatial accuracy can be quantified by spatial characteristics, such as contrast and texture between the predicted and the true images. For spatial evaluation metrics, Robert’s edge (Edge) was used to describe the spatial accuracy of the predicted images [75]. A value closer to 0 indicates a better image fusion result; a negative value indicates that the edge features are smoothed, and a positive value indicates that the edge features are sharpened. Table 3 shows the equations of accuracy metrics and the meaning of each variable.

## 4. Results

### 4.1. Qualitative Comparison

To compare the proposed method with the original ESTARFM method, the same input was used for both algorithms. Three pairs of Landsat8 OLI and MODIS images were acquired on January, February, and March. Figure 2 shows the images using RGB composites. The two pairs of Landsat8 OLI and MODIS images acquired in January and March were used to predict the image at the Landsat spatial resolution in February. Then, the predicted image was compared with an actual Landsat image acquired in February to evaluate the performance of the two methods. Figure 7 shows the images predicted by the ESTARFM and iESTARFM, respectively. Figure 7a shows the actual Landsat image, where the major land cover types are snow and land located in high-altitude mountain areas. At high latitudes, large areas are covered by snow with high reflectance in visible bands. As the altitude decreases, the amount of snow gradually decreases. Figure 7b,c present the predicted results of the ESTARFM and iESTARFM methods. A zoom-in area in the first and third rows is used to highlight the details between the predicted image and the actual image.

From the visual comparison of the overall image, the predicted snow cover boundaries of the two methods are similar to those of the actual Landsat image in Figure 7, indicating that the two methods can capture the major changes when the snow melts. However, both methods have common limitations, such as a blurring boundary between the snow-covered area and the land area and they also miss some tiny parts of the snow. Figure 7b presents the predicted image of ESTARFM, where bright abnormal patches are generated on the land area due to the wrong selection of similar pixels. The Figure 7c shows the predicted results of iESTARFM, where the over-bright patches and noise are reduced, and clearer texture structures are obtained. Furthermore, comparing the zoom-in area in the first row of the two methods in Figure 7b, it can be seen that bright noise is generated by the ESTARFM method, while iESTARFM can reduce such noise, and the predicted result is closer to the actual image. The third row of Figure 7 presents the areas with significant changes in snow melt. It can be seen that the ESTARFM cannot accurately predict the pixel values which the non-snow pixels are incorrectly predicted as snow. In contrast, iESTARFM is better at predicting the values. The visual comparison results indicate that the image predicted by iESTARFM is more similar to the actual image in terms of spatial details.

### 4.2. Quantitative Comparison

To further verify the effectiveness of the proposed method, eight accuracy metrics were used to evaluate the two methods, and the metrics are presented in Table 3. MSE, RMSE, r, ERGAS, SAM, PSNR, and SSIM were selected to evaluate the spectral accuracy, and they are sensitive to errors. EDGE was selected to evaluate spatial accuracy. The set of accuracy metrics was calculated for the two predicted results, where better prediction results are marked in bold, as shown in Table 4. The iESTARFM method performed better among the average of 6 bands of the evaluation metrics. To show the accuracy of evaluation results more clearly, a bar chart is used to show comparison results in Figure 8.

In the comparison of spectral accuracy, the iESTARFM provided the most accurate predictions with the smallest MSE, RMSE, and ERGAS, as well as the highest r, except for the SWIR2 band. The MSE and RMSE indicate the deviation between the true and predicted values. The prediction results by iESTARFM have smaller accuracy values in the six bands, where the mean value of MSE is 0.007 (reduced by 0.003 as compared to 0.010) and the mean value of RMSE is 0.078 (reduced by 0.017 as compared to 0.095), indicating that the spectral values of iESTARFM are closer to the true value. r reflects the linear correlation between the predicted and true values, and a value closer to 1 indicates a higher correlation. The accuracy of iESTARFM is greater than 0.9 in the visible and near-infrared bands, but less than 0.9 in the two short-wave infrared bands. The mean value of r is 0.899 in the six bands (increased by 0.013 as compared to 0.886), indicating that iESTARFM has a higher correlation in visible and near-infrared bands but a lower correlation in shortwave infrared bands. To describe the linear relationship more clearly, the scatter plots of the predicted and true values for each band of ESTARFM and iESTARFM are illustrated in Figure 9 and Figure 10. The scatter plots of ESTARFM are more discrete, and those of iESTARFM are more aggregated, indicating that the predictions of iESTARFM are closer to the actual values than the ESTARFM method. A smaller ERGAS value indicates a higher fidelity of the prediction results. The ERGAS of iESTARFM is less than 6.4 in all six bands, and that of ESTARFM is greater than 7.0. The mean value of iESTARFM is 5.131 (reduced by 3.081 as compared to 8.212), which indicates that iESTARFM has a better texture performance. SAM was used to calculate the similarity between two images. A smaller SAM value indicates a higher similarity between two images. The iESTARFM has a smaller SAM value of less than 15.0 in the six bands with a mean value of 11.995, and the SAM value of ESTARFM is greater than 18.0, indicating that iESTARFM has better reconstruction results. A larger PSNR value indicates better image quality. The mean PSNR value of iESTARFM is 21.275 and that of ESTARFM is 19.736 among the six bands, indicating that iESTARFM can obtain better image quality. The SSIM value closer to 1 indicates that the two images are more similar. The SSIM value of iESTARFM is greater than 0.9 in the visible and near-infrared bands, with a mean value of 0.896. iESTARFM has better accuracy than ESTARFM except for the SWIR2 band, and the prediction result has a higher structural similarity to the true value. In the comparison of spatial accuracy, the EGDE value closer to 0 indicates that the predicted image texture features are more similar to the true value. The Edge values of ESTARFM and iESTARFM are less than 0 in the six bands, indicating that the predicted image of both methods shows a smooth effect compared to the true image. The Edge value of iESTARFM is less than −0.299 and that of ESTARFM is greater than −0.299 in the visible and near-infrared bands, and for the two SWIR bands, ESTARFM has a smaller Edge value than iESTARFM. The mean Edge value of ESTARFM is −0.29, and that of iESTARFM is −0.264. Generally, iESTARFM can obtain more similar texture features.

## 5. Discussion

The change in snow cover significantly affects the exchange of energy between the atmosphere and land surface, which is important for theoretical studies and practical applications. However, due to technical and environmental limitations, it is hard to obtain images with both high spatial and temporal resolution just using a single satellite [16], especially in the mountain region, where the spatial and temporal variability of the snow-cover is particularly high [17]. The ESTARFM assumes no abrupt changes in the surface type, which limits its application in snow-covered mountain areas. In fact, the surface type may change abruptly due to snow and snow melting. This may cause problems with ESTARFM in the selection of similar pixels. The threshold of selecting similar pixels will be higher due to the presence of snow, which makes the wrong selection of pixels. Thus, the iESTARFM method has made some improvements based on the original ESTARFM method, considering the optical characteristics of snow and the high correlation between the change in snow cover and the elevation. The main idea of iESTARFM is to introduce NDSI and DEM to simulate the change in snow cover. By qualitative and quantitative comparison, the experiment found that iESTARFM has higher accuracy than ESTARFM

Although iESTARFM can predict good results, there are still several limitations and constraints. Firstly, we calculated the NDSI of the base images. The NDSI threshold is based on the histogram statistical method and surface reflectance. Although it can be used effectively and accurately to estimate snow cover information from satellite images, the disadvantage is that this approach is relatively subjective. The closer the snow-cover is to the real surface, the more accurate the selection of similar pixels. So, it is necessary to explore more accurate snow-cover mapping methods, such as the decision-tree-based classification model [76], the supervised fuzzy classification approach [77] and subpixel snow-cover mapping for automated mapping of snow cover. Secondly, we use DEM to simulate the snow cover on the target date. Although we can infer the change in snow cover from elevation, the simulation accuracy is still limited because there are many factors related to the snow-covered change, such as daytime air temperature, distance to significant open water bodies, topographic roughness and aspect, forest cover, and snow class. There is blurring at the boundaries of the snow-covered area because snow covers cannot be accurately identified based on NDSI and DEM. Finally, iESTARFM performs well in the visible and near-infrared bands, but in two shortwave infrared bands, the r, SSIM, and EDGE values are lower than those of ESTARFM. This may be because compared to non-snow pixels, the snow pixels have a particularly high reflectance in the visible and near-infrared bands, which can be easily distinguished. Furthermore, it is difficult to separate glacier from snow using optical remote sensing images.

In the future, there will still be much work to do to improve the accuracy of our method. Firstly, other useful information such as temperature data, slope, and aspect can be considered to obtain the snow change information. Secondly, the datasets with a higher resolution can be considered to capture more details of ground objects. Thirdly, it is necessary to find other study areas. Three conditions need to be met to select the study area. First, MODIS and Landsat images should be at the same time and place at high altitude. Second, the images should be largely free of clouds and shadows. Thirdly, there is a gradual trend of snow-covered change. Thus, the reliability of the method does need to be explored in more study areas. We will experiment with more similar study areas to explore the applicability of our approach and compare it with more spatiotemporal algorithms, as well as consider the advantages of other algorithms to improve the accuracy of data fusion not just in the high-altitude snow areas.

## 6. Conclusions

The spatial and temporal distribution of snow is important to the changes in climate. However, due to the limitations of technology and atmospheric conditions, it is hard to obtain images with both high spatial and temporal resolution just using a single satellite [16], especially in mountain regions. Thus, the iESTARFM is developed for snow-covered mountain areas. The main idea of this method is to improve the accuracy of selecting similar pixels by introducing NDSI and DEM information that can simulate the change in snow cover. There are three main steps. Firstly, simulate snow-cover changes using NDSI and DEM information. Secondly, select similar pixels according to the results of snow-covered changes. Thirdly, the thresholds are calculated separately according to the pixels type. The prediction results of ESTARFM and iESTARFM methods are evaluated qualitatively and quantitatively. For the visual evaluation, both algorithms can simulate snow cover boundaries. However, the ESTARFM method could generate bright abnormal patches in the land area due to the wrong selection of similar pixels, while the iESTARFM made good predictions in the land area. For the quantitative analysis, eight evaluation metrics commonly used for the spatiotemporal fusion method were selected. The iESTARFM has better accuracy than ESTARFM in the visible, NIR, and SWIR bands. In addition to SWIR1 in r and EDGE evaluation metrics, SWIR2 in SSIM evaluation metric, ESTARFM has a better performance than iESTARFM. From the scatter plots iESTARFM is more focused and has a higher correlation coefficient. The correlation coefficients are greater than 0.9 in the visible and NIR bands, and less than 0.87 in short-wave infrared bands. Because snow has higher reflectance in the visible and near-infrared bands than short-wave infrared bands, the values have a larger distribution from 0 to 1 in the visible bands, and the values between 0 and 0.5 in the short-wave infrared bands. The evaluation metrics perform well in the visible and near-infrared bands, probably because the reflectance characteristics of snow are more distinguished compared to other objects in the visible and near-infrared bands. In the future, this method could be used to generate dense time series images for snow-covered mountain areas.

## Figures and Tables

**Figure 1 sensors-22-08524-f001:**
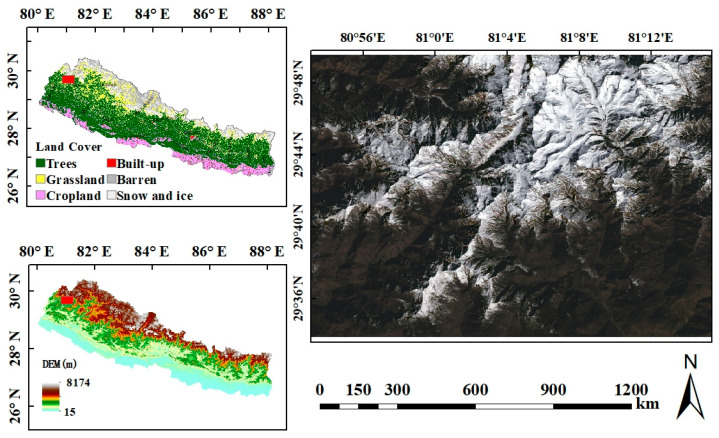
The location map of the study area. The top left image shows the land cover types of Nepal. The bottom left image shows the DEM of Nepal. The right image shows the RGB composites of the Landsat surface reflectance image on 12 February 2020.

**Figure 2 sensors-22-08524-f002:**
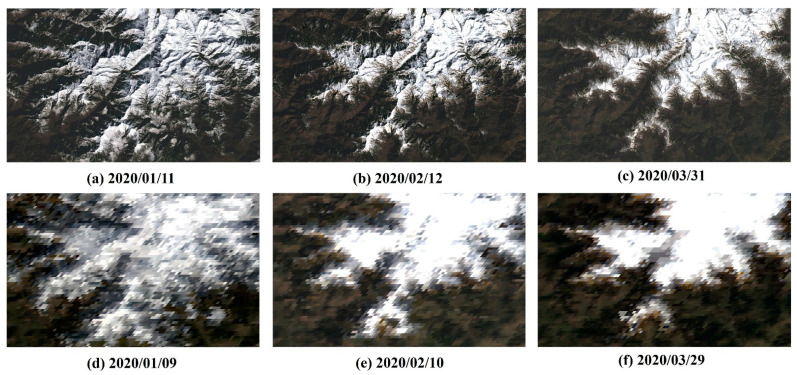
The RGB composites of MODIS and Landsat surface reflectance. (**a**–**c**) are Landsat surface reflectance images on 2020/01/11, 2020/02/12, and 2020/03/31, and the percentage of snow cover is 39.07%, 25.44%, and 22.71%. (**d**–**f**) are MODIS surface reflectance images on 2020/01/09, 2020/02/10, and 2020/03/29, and the percentage of snow cover is 48.44%, 30.72%, and 25.30%, respectively.

**Figure 3 sensors-22-08524-f003:**
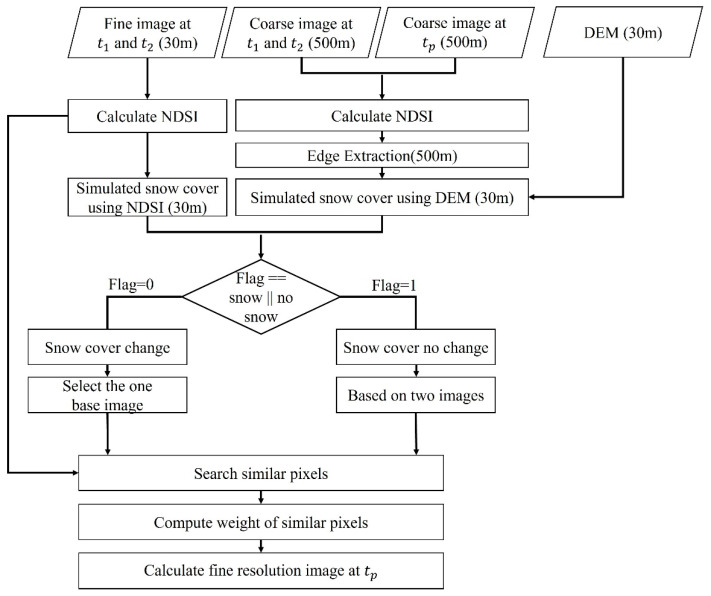
The flowchart of the iESTARFM algorithm.

**Figure 4 sensors-22-08524-f004:**
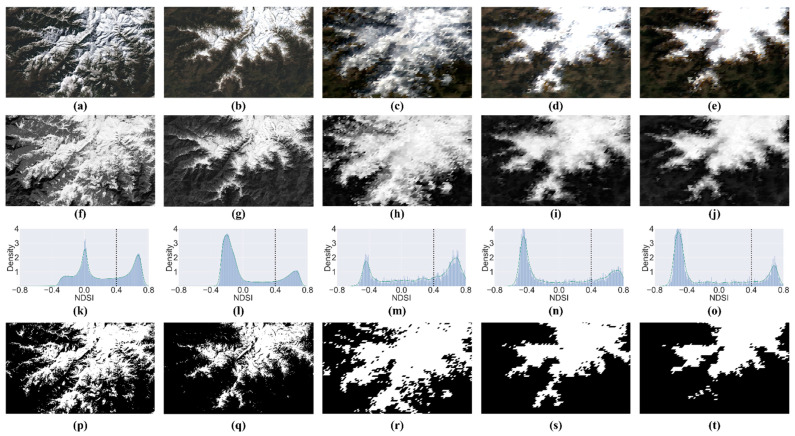
The process of generating the snow mask by using NDSI. Columns 1 and 2 show Landsat images at $t_{1}$ and $t_{2}$. Columns 3, 4, and 5 show MODIS images at $t_{1}$, $t_{p}$, and $t_{2}$. (**a**–**e**) are the surface reflectance images; (**f**–**j**) are the NDSI images; (**k**–**o**) are the NDSI frequency histogram; (**p**–**t**) are the snow mask based on a threshold of 0.4.

**Figure 5 sensors-22-08524-f005:**
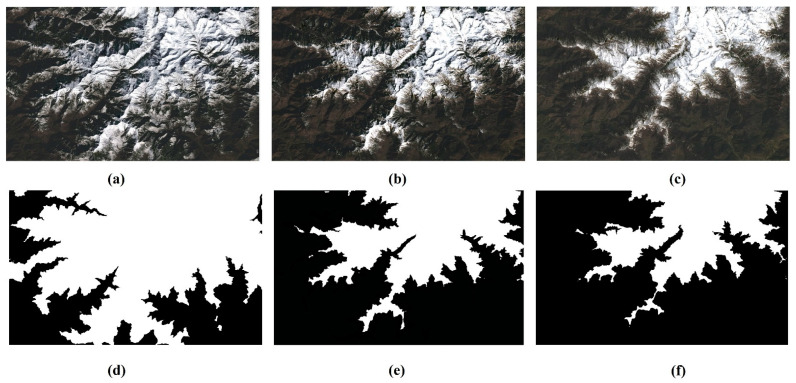
The Landsat surface reflectance image and snow mask using DEM. (**a**–**c**) are the true Landsat surface reflectance images on 2020/01/11, 2020/02/12, and 2020/03/31. (**d**–**f**) are the corresponding snow masks by setting different thresholds of DEM data. (**d**) A height of 3000 m is used as the threshold. (**e**) A height of 3600 m is used as the threshold. (**f**) A height of 3700 m is used as the threshold.

**Figure 6 sensors-22-08524-f006:**
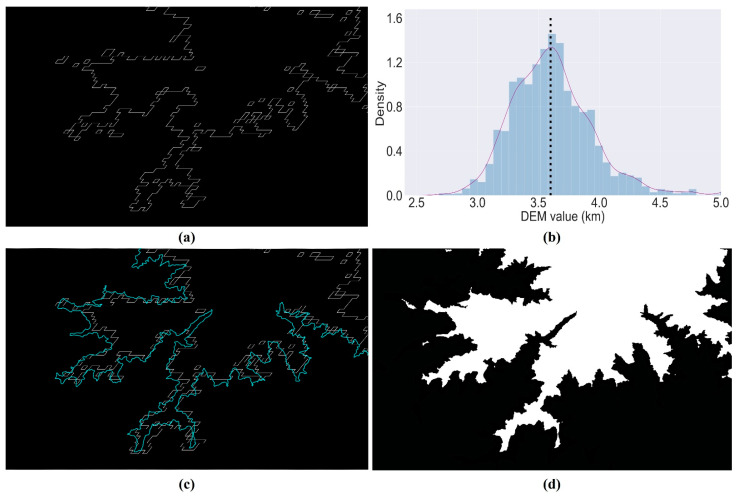
The process of making snow masks using DEM. (**a**) is the snow cover boundary extracted using MODIS NDSI. (**b**) is the frequency histogram of the DEM data located at the extracted snow cover boundary from MODIS NDSI. (**c**) The bright line is the extracted boundary overlaid on the DEM height of 3600 m. (**d**) is the snow mask extracted from DEM. The white part is snow, and the black part is non-snow.

**Figure 7 sensors-22-08524-f007:**
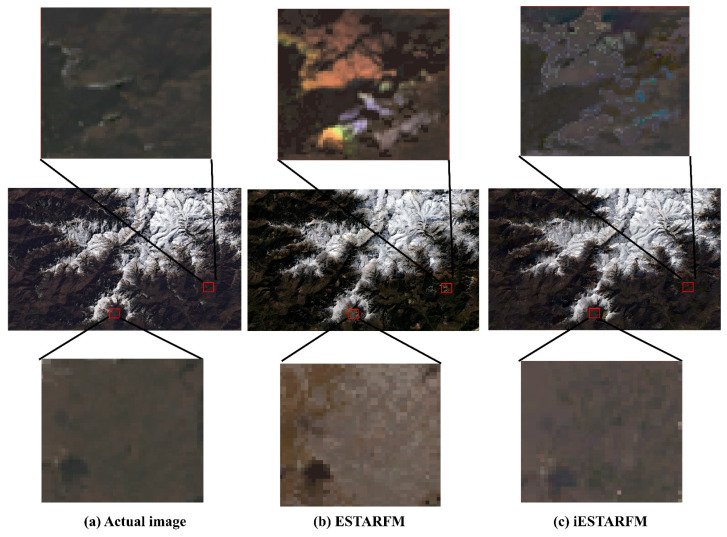
Comparison of the actual and predicted images. (**a**) is the actual image observed on 12 February 2020. (**b**) is the prediction image by ESTARFM. (**c**) is the prediction image by iESTARFM. The images of the first and third rows are the zoom-in areas of three images.

**Figure 8 sensors-22-08524-f008:**
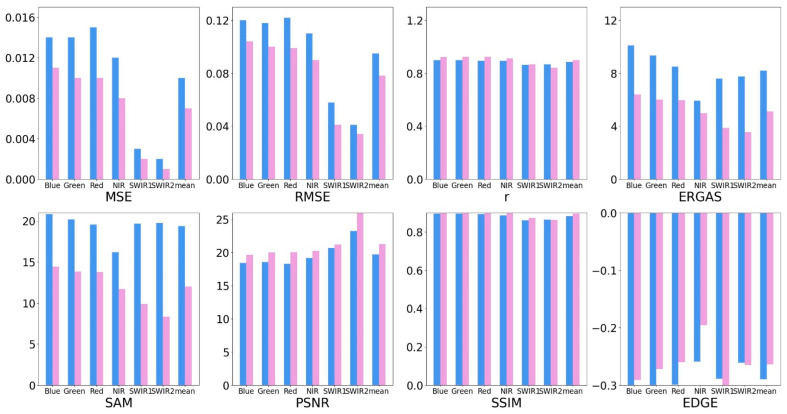
The accuracy evaluation results for ESTARFM and iESTARFM. The blue bars represent the result of the ESTARFM method, and the pink bars represent the result of the iESTARFM method.

**Figure 9 sensors-22-08524-f009:**
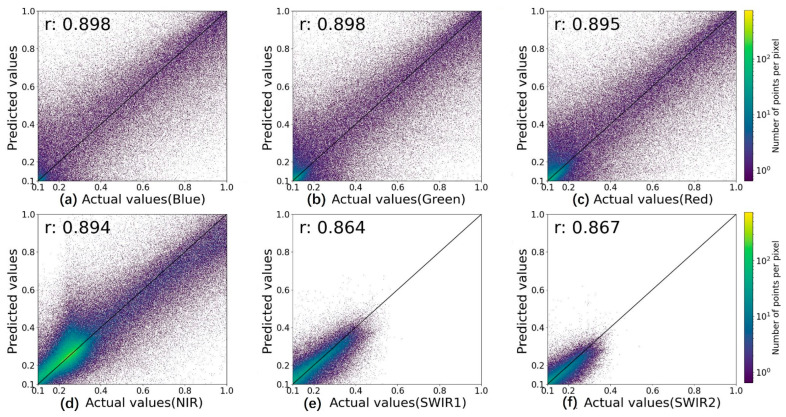
Scatter plots of the actual and predicted values for the six bands of ESTARFM. ((**a**–**f**) for different bands and the dark line is a 1:1 line).

**Figure 10 sensors-22-08524-f010:**
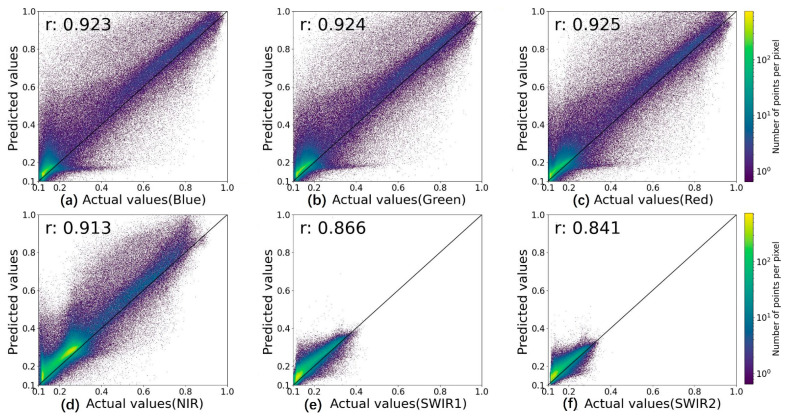
Scatter plots of the actual and predicted values for the six bands of iESTARFM. ((**a**–**f**) for different bands and the dark line is a 1:1 line).

**Table 1 sensors-22-08524-t001:** Information of corresponding bands of Landsat8 OLI, and MODIS.

Band	Landsat 8 OLI	Bandwidth (nm)	MODIS	Bandwidth (nm)
Blue	Band 2	450–510	Band 3	459–479
Green	Band 3	530–590	Band 4	545–565
Red	Band 4	630–690	Band 1	620–670
Near Infrared	Band 5	850–880	Band 2	841–876
Short-Wave Infrared 1 (SWIR1)	Band 6	1570–1650	Band 6	1628–1652
Short-Wave Infrared 2 (SWIR2)	Band 7	2110–2290	Band 7	2105–2155

**Table 2 sensors-22-08524-t002:** Remote sensing data types and acquisition date.

Data Type	Spatial Resolution	Temporal Resolution	Acquisition Date	Expression	Use	Percentage of Snow	Percentage of Cloud
Landsat8 OLI	30 m	16 days	2020/01/11	$F_{t_{1}}$	Base Image	39.07	<5
	30 m		2020/02/12	$F_{t_{P}}$	Evaluation	25.44	<5
	30 m		2020/03/31	$F_{t_{2}}$	Base Image	22.71	<5
MODIS	500 m	daily	2020/01/09	$C_{t_{1}}$	Base Image	48.44	<5
	500 m		2020/02/10	$C_{t_{P}}$	Base Image	30.72	<5
	500 m		2020/03/29	$C_{t_{2}}$	Base Image	25.30	<5

**Table 3 sensors-22-08524-t003:** Equations for accuracy metrics and the meaning of each variable.

Metric Name	Equation	Variable Explanation
MSE	$\;MSE=\frac{1}{N}\sum_{i=1}^{N}{\left(F_{i}-R_{i}\right)}^{2}$	$R_{i}$: the value of the *i*th pixel in the true image
RMSE	$\;RMSE=\sqrt{\frac{\sum_{i=1}^{N}{\left(F_{i}-R_{i}\right)}^{2}}{N}}$	$F_{i}$: the value of the *i*th pixel in the predicted image
r	$\;r=\frac{\sum_{i=1}^{N}\left(R_{i}-\mu_{R}\right)\left(F_{i}-\mu_{F}\right)}{\left(\sqrt{\sum_{i=1}^{N}{\left(R_{i}-\mu_{R}\right)}^{2}}\right)\left(\sqrt{\sum_{i=1}^{N}{\left(F_{i}-\mu_{F}\right)}^{2}}\right)}$	$\mu_{F}$ the mean pixel values of the predicted image
ERGAS	$\;ERGAS=100\sqrt{\frac{1}{B}\overset{B}{\underset{j=1}{\sum }}{\left(\frac{RMSE\left(F_{i}\right)}{\mu_{j}}\right)}^{2}}$	$\mu_{R}$: the mean pixel values of the true image
SSIM	$\;SSIM=\frac{\left(2\mu_{R}\mu_{F}+c_{1}\right)\left(2\mu_{RF}+c_{2}\right)}{\left(\mu_{R}{}^{2}+\mu_{F}{}^{2}+c_{1}\right)\left(\sigma_{R}{}^{2}+\sigma_{F}{}^{2}+c_{2}\right)}$	$\sigma_{F}$: the variance of pixel values of the predicted image
SAM	$\;SAM=arccos\frac{\sum \left(F_{i}R_{i}\right)}{\sqrt{R_{i}{}^{2}F_{i}{}^{2}}}$	$\sigma_{R}$: the variance of pixel values of the true image
PSNR	$PSNR\;=\;10{log}_{10}\frac{255^{2}}{MSE}$	$c_{1},c_{2}$: constants N: the total number of pixels
Edge	$Edge=\left|D_{i,j}-D_{i+1,j+1}\right|+\left|D_{i,j+1}-D_{i+1,j}\right|$ $Edge\;error=R_{Edge}-F_{Edge}$	D: the value of *i*th pixel in the moving window

**Table 4 sensors-22-08524-t004:** Accuracy assessment of the ESTARFM and iESTARFM methods.

Metrics	Method	Band						
		Blue	Green	Red	NIR	SWIR1	SWIR2	Average
MSE	ESTARFM	0.014	0.014	0.015	0.012	0.003	0.002	0.010
	iESTARFM	**0.011**	**0.010**	**0.010**	**0.008**	**0.002**	**0.001**	**0.007**
RMSE	ESTARFM	0.120	0.118	0.122	0.110	0.058	0.041	0.095
	iESTARFM	**0.104**	**0.100**	**0.099**	**0.090**	**0.041**	**0.034**	**0.078**
r	ESTARFM	0.898	0.898	0.895	0.894	0.864	0.867	0.886
	iESTARFM	**0.923**	**0.924**	**0.925**	**0.913**	**0.866**	**0.841**	**0.899**
ERGAS	ESTARFM	10.113	9.348	8.505	5.935	7.605	7.764	8.212
	iESTARFM	**6.389**	**6.013**	**5.964**	**4.985**	**3.881**	**3.555**	**5.131**
SAM	ESTARFM	20.825	20.195	19.567	16.188	19.675	19.75	19.367
	iESTARFM	**14.428**	**13.827**	**13.763**	**11.717**	**9.885**	**8.351**	**11.995**
PSNR	ESTARFM	18.422	18.569	18.291	19.162	20.708	23.263	19.736
	iESTARFM	**19.634**	**19.997**	**20.045**	**20.256**	**21.226**	**26.494**	**21.275**
SSIM	ESTARFM	0.896	0.896	0.893	0.887	0.861	0.865	0.883
	iESTARFM	**0.911**	**0.914**	**0.915**	**0.902**	**0.872**	**0.863**	**0.896**
EDGE	ESTARFM	−0.328	−0.307	−0.299	−0.259	−0.289	−0.261	−0.29
	iESTARFM	**−0.291**	**−0.272**	**−0.260**	**−0.196**	**−0.3**	**−0.265**	**−0.264**

## Data Availability

Not applicable.

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
