# Peer review of "An Improved Spatiotemporal Data Fusion Method for Snow-Covered Mountain Areas Using Snow Index and Elevation Information"

_sensors, 2022, doi:10.3390/s22218524_

Round 1

Reviewer 1 Report

See attached review.

Author Response

We appreciate the time and efforts by you and your referees in reviewing this manuscript. We have studied the reviewers’ comments and recommendations carefully and tried our best to revise our manuscript accordingly. For easier reading, our responses are written in red color and detail changes in the "Track Changes" version of the revised manuscript are also written in red color.

Since we can only upload one PDF, we combined the point-by-point responses and the modified manuscript in one PDF.

Reviewer 2 Report

The article tries to improve the imaging and monitoring the snow regions. I think the article is important for remote sensing, appropriate for the journal topics and can be published after minor corrections.

1) The references do not contain full information: nor DOI, nor authors.

2) There are misprints like "Figure 1. shows". There are unclear symbols "错误!未找 212 到引用源" (and in other places). See lines 448-450.

3) Figures' axis captions are often too small to read (especially, in fig. 9-10, but not only).

4) The paragraphs are so huge that it is difficult to understand the idea of each one.

5) It is unclear from the discussion how reliable the technique is, how it agrees with other techniques, and what its limits are.

6) I would like to know the false negative and false positive errors when you predict snow at the pixel and there is no and when you predict no snow, but there is.

7) The title is too difficult to understand - "An improved Spatiotemporal Data Fusion Method for Snow-covered Mountain Areas Using NDSI and DEM Information based on ESTARFM"

- there are too many abbreviations. Who will know in 10-20 years what NDSI, DEM or ESTARFM mean?

- there is uncertainty about where is the main word for "based"

- I would suggest "spatio-temporal" instead of "spatiotemporal"

Author Response

(The authors gave the same response as above.)
